# Endothelial Dysfunction and Pregnant COVID-19 Patients with Thrombophilia: A Narrative Review

**DOI:** 10.3390/biomedicines11092458

**Published:** 2023-09-04

**Authors:** Metodija Sekulovski, Niya Mileva, Lyubomir Chervenkov, Monika Peshevska-Sekulovska, Georgi Vasilev Vasilev, Georgi Hristov Vasilev, Dimitrina Miteva, Latchezar Tomov, Snezhina Lazova, Milena Gulinac, Tsvetelina Velikova

**Affiliations:** 1Department of Anesthesiology and Intensive Care, University Hospital Lozenetz, 1 Kozyak Str., 1407 Sofia, Bulgaria; 2Medical Faculty, Sofia University, St. Kliment Ohridski, Kozyak 1 Str., 1407 Sofia, Bulgaria; mpesevska93@gmail.com (M.P.-S.); vvasilev.georgi@gmail.com (G.V.V.); drgeorgivasilev@gmail.com (G.H.V.); d.georgieva@biofac.uni-sofia.bg (D.M.); lptomov@nbu.bg (L.T.); snejina@lazova.com (S.L.); mgulinac@hotmail.com (M.G.); tsvelikova@medfac.mu-sofia.bg (T.V.); 3Medical Faculty, Medical University of Sofia, 1 Georgi Sofiiski Str., 1431 Sofia, Bulgaria; nmileva91@gmail.com; 4Department of Diagnostic Imaging, Medical University Plovdiv, Bul. Vasil Aprilov 15A, 4000 Plovdiv, Bulgaria; lyubo.ch@gmail.com; 5Department of Gastroenterology, University Hospital Lozenetz, 1407 Sofia, Bulgaria; 6Clinic of Endocrinology and Metabolic Disorders, UMHAT “Sv. Georgi”, 4000 Plovdiv, Bulgaria; 7Laboratory of Hematopathology and Immunology, National Specialized Hospital for Active Treatment of Hematological Diseases, “Plovdivsko Pole“ Str., 6, 1756 Sofia, Bulgaria; 8Department of Genetics, Faculty of Biology, Sofia University “St. Kliment Ohridski”, 8 Dragan Tzankov Str., 1164 Sofia, Bulgaria; 9Department of Informatics, New Bulgarian University, Montevideo 21 Str., 1618 Sofia, Bulgaria; 10Pediatric Clinic, University Hospital “N. I. Pirogov,” 21 “General Eduard I. Totleben” Blvd; 1606 Sofia, Bulgaria; 11Department of Healthcare, Faculty of Public Health “Prof. Tsekomir Vodenicharov, MD, DSc”, Medical University of Sofia, Bialo More 8 Str., 1527 Sofia, Bulgaria; 12Department of General and Clinical Pathology, Medical University of Plovdiv, Bul. Vasil Aprilov 15A, 4000 Plovdiv, Bulgaria

**Keywords:** endothelial dysfunction, pregnant women, pregnancy, COVID-19, SARS-CoV-2 infection, thrombophilia

## Abstract

Pregnancy with SARS-CoV-2 infection can raise the risk of many complications, including severe COVID-19 and maternal–fetal adverse outcomes. Additionally, endothelial damage occurs as a result of direct SARS-CoV-2 infection, as well as immune system, cardiovascular, and thrombo-inflammatory reactions. In this narrative review, we focus on endothelial dysfunction (ED) in pregnancy, associated with obstetric complications, such as preeclampsia, fetal growth retardation, gestational diabetes, etc., and SARS-CoV-2 infection in pregnant women that can cause ED itself and overlap with other pregnancy complications. We also discuss some shared mechanisms of SARS-CoV-2 pathophysiology and ED.

## 1. Introduction

COVID-19 in pregnancy is still a severe risk factor for complications in both maternal (i.e., need for intensive care, death) and fetal aspects (i.e., miscarriages, preeclampsia, intrauterine growth retardation, preterm labor, stillbirth, etc.) [1], although the World Health Organisation has declared the pandemic is no longer a global concern. The higher risk of pregnancy complications associated with SARS-CoV-2 infection is probably related to cardiovascular, hormonal, and immunological amendments in pregnant women, changes that make females prone to more severe COVID-19 [1,2,3].

Some research groups reported a higher incidence of preeclampsia in SARS-CoV-2-infected women [4,5]. In addition to COVID-19, other pandemics, such as H1N1, MERS, and SARS-CoV-1, have had catastrophic consequences on pregnancy outcomes. Preterm birth, preeclampsia, preterm prelabor membrane rupture, fetal development restriction, and mode of delivery were among the ones that were noted. Fetal distress, an Apgar score below seven at five minutes, neonatal asphyxia, admission to a neonatal intensive care unit, perinatal death, and signs of vertical transmission were the perinatal outcomes seen [6]. However, other researchers questioned that incidence, suggesting careful analysis and interpretation of the available data [7].

Furthermore, it is thought that extensive endothelial dysfunction (ED) is the common and overlapping pathophysiological mechanism behind both preeclampsia and COVID-19 [8,9,10]. SARS-CoV-2 infection and preeclampsia exert symptoms related to vasoconstriction and organ ischemia. When both conditions are present, it is difficult to determine which complications are attributed to one or the other condition, taking into account that ED is the common feature. Usually, ED is presented by elevated levels of angiotensin II (Ang-II) and the antiangiogenic sFlt-1 (FMS-like tyrosine kinase 1) and low levels of angiogenic molecule placental growth factor (PIGF). However, a high sFlt-1/PIGF ratio could propose placental dysfunction in preeclampsia, but excessive levels are observed in SARS-CoV-2-infected pregnant women [9,10].

In this review, we focus on ED in pregnancy associated with obstetric complications, such as preeclampsia, fetal growth retardation, gestational diabetes, etc., and SARS-CoV-2 infection in pregnant women that can cause ED and overlap with other pregnancy complications.

## 2. Search Strategy

We conducted a search through scientific—bibliographic— databases, Medline (PubMed) and Scopus. We used MeSH and free-text terms: (“COVID-19” OR “SARS-CoV-2”) AND (“pregnant” OR “pregnancy”) AND (“endothelial dysfunction”) AND (“thrombophilia”). Relevant papers were included in a modified form of a narrative review [11].

## 3. Endothelial Dysfunction during COVID-19 in Pregnancy—Cardiology Point of View

Although the endothelium represents a simple cell monolayer, its crucial location as a vessel’s inner layer determines its essential role. Healthy endothelium can respond to physical and chemical signals by producing a wide range of factors that regulate vascular tone, cellular adhesion, thromboresistance, smooth muscle cell proliferation, and vessel wall inflammation. The endothelium is the key regulator of vascular homeostasis, as it determines vascular tone, smooth muscle cell proliferation, vessel wall inflammation, and platelet aggregation [12]. An alteration in endothelial function results in the disproportionate production of relaxing and contracting factors and procoagulants. ED precedes the development of atherosclerotic changes. It is increasingly recognized as an important prognostic factor for cardiovascular events [13].

Endothelial function assessment can enhance risk stratification, improve early diagnosis, and be used to assess therapeutic response [14]. The optimal method for endothelial function evaluation should be safe, reproducible, and standardized between laboratories. Both invasive and noninvasive tests have been developed. However, some challenges arise when endothelial function has to be assessed in pregnant women.

### 3.1. Invasive Testing

In general, endothelial function assessment is mainly focused on vasoreactivity testing, as this is the most verifiable function of the vascular endothelium [15]. Assessing endothelial function in the cardiac catheterization laboratory is feasible in patients with risk factors for atherosclerotic coronary artery disease [12].

The intracoronary acetylcholine provocation test is the gold standard for vasoreactivity testing [16]. Acetylcholine binds muscarinic receptors, which leads to nitric oxide (NO) release and subsequent arterial dilatation in the case of an intact endothelium. However, acetylcholine induces arterial conduit constriction when there is an endothelium impairment. Dosing regimens for acetylcholine during provocation testing are diverse, with some doses starting at 2 µg to up to 200 µg. Notably, the current data supporting the dosing regimens of acetylcholine during provocation testing are mainly focused on coronary epicardial and microvascular vasospasm and not on endothelial dysfunction testing. ED is diagnosed when the reproduction of symptoms and ST-segment depression on ECG are apparent [17].

Additionally, the endothelial function can be invasively assessed by measuring coronary blood flow changes and, thereby, coronary flow reserve using a Doppler wire [15] or using the method of thermodilution [18]. A Doppler-tipped guidewire is most commonly positioned in a proximal segment of a coronary artery (usually at the left anterior descending coronary artery), and the Doppler flow velocity is continuously recorded. The change in blood flow velocity at rest and after provocation with acetylcholine is calculated as a measure of the preserved or impaired ED. Although the described invasive tests are characterized by high specificity and sensitivity, they require specific technical equipment and experienced personnel. Hence, provocative invasive testing is not performed regularly [19].

### 3.2. Noninvasive Testing

The application of noninvasive tests allows for accessible and easily applicable measurements of endothelial function in everyday clinical practice. The brachial artery is the most accessible location to assess vasoreactivity, specifically its flow-mediated dilatation. The latter can be achieved through B-mode ultrasound and Doppler, evaluating flow changes in response to vasomotor stimuli such as blood pressure cuff inflation or oral/sublingual nitroglycerin [20]. Various factors, including temperature, food, caffeine, and drugs, may influence vascular reactivity. Therefore, subjects should fast for at least 12 h before the examination, and the test should be performed in a quiet, temperature-controlled room. To create a flow stimulus in the brachial artery, a sphygmomanometric cuff is positioned above the antecubital fossa. A baseline rest image is acquired, and blood flow is estimated by time-averaging the pulsed Doppler velocity signal obtained. Afterward, ischemia is induced by the arterial occlusion created from the cuff inflation to supra systolic pressure. The following cuff deflation induces a high-flow state, which increases shear stress and brachial artery dilatation. After cuff deflation, a control image and Doppler signal are obtained to assess hyperemic velocity.

Other noninvasive modalities allowing for the reproducible and safe evaluation of endothelial function are magnetic resonance imaging (MRI) and positron emission tomography (PET). Both methods provide the ability to quantify coronary blood flow. To evaluate coronary blood flow and myocardial flow reserve (MFR), different provoking tests, such as isometric handgrip exercise [21] or cold pressor tests, have been validated [22]. MFR is calculated as the ratio of stress to rest blood flow, and a value below 2.0 is considered abnormal and consistent with ED [23].

The decision for the optimal test should be made after taking into consideration each specific patient’s characteristics and pathology. In the case of pregnancy, we should opt for the minimally invasive test, ideally without any provocative medication application.

Flow-mediated dilatation (FMD) and its modified version-reactive hyperemial arterial tonometry [24,25,26] are the gold standard for evaluating ED in preeclampsia among noninvasive methods.

It was shown that a severe ED was observed in pregnant women with preeclampsia, manifested with reduced FMD, compared to healthy pregnant women [24]. Usually, these differences are more prominent in the third trimester or around the 30th gestational week. However, ED will have developed weeks before the clinical manifestation of preeclampsia [27].

Imaging plays a crucial role in the diagnosis of blood vessel diseases. Diagnostics in this field has come a long way from the use of classical angiography through DSA (digital subtraction angiography), which nowadays is mainly used in invasive procedures, to computed tomography angiography (CTA) and magnetic resonance angiography (MRA). Molecular magnetic resonance is a modern method for diagnosing vascular diseases, especially for evaluating the endothelium.

Direct in vivo study of the vascular endothelium is complex. Attempts have been made to diagnose the endothelium using physical methods, and various magnetic resonance sequences have been developed in recent years [28]. By applying gadolinium, the contrast of the vessel wall can be evaluated, which is a semiquantitative analysis of vascular permeability. One of the ways to apply contrast material is through so-called dynamic contrast enhancement (DCE), which allows evaluation of the contrast of the vascular wall in an active order. It is possible to measure the microvascular volume, contrast extravasation rate, time to peak, and peak concentration [28]. Performing a quality dynamic contrast study is challenging, even with modern devices, and requires good spatial and temporal resolution because the vessel wall is a very fine structure. A group of scientists proposes using a specific method called a quantitative MRI of endothelial permeability and dysfunction (qMETRIC). The method is a specially developed protocol that evaluates late gadolinium enhancement (LGE) and modified Look –Locker inversion recovery (MOLLI) T1. It is currently under investigation using mice but has the potential to be a useful modern biomarker assessing vascular permeability. This way, modern diagnostics is performed noninvasively and relatively quickly and is affordable. The large vessels, such as the aorta, aortic arch, and brachiocephalic and carotid arteries, are suitable for evaluation. The disadvantage is the need for a special coil and exact patient positioning. The examination must be carried out with ECG monitoring, which must also be precise [29].

However, one must acknowledge that searching for proper and valuable biomarkers for endothelial dysfunction could be challenging. Moreover, a sophisticated biomarker could be difficult to implement in clinical practice. However, it could be helpful in the research.

In recent years, different sequences have been developed to study the endothelial function of coronary vessels, one of which is the late gadolinium enhancement. Thanks to this sequence, the assessment of endothelium-dependent coronary vasomotor abnormality is possible [30].

## 4. Endothelial Dysfunction and COVID-19

At the beginning of the COVID-19 pandemic, the lungs’ involvement in infected patients stood out. Most studies have focused on understanding the mechanisms of lung damage, leading to the discovery that the virus enters the lungs via ACE2 receptors. However, the ongoing pandemic has highlighted other complications in infected patients. In addition to pulmonary involvement, ischemic brain infarctions, renal failure, heart attacks and myocardial damage, pulmonary thromboembolism, venous thrombosis, and, less often, gastrointestinal tract and liver diseases were found in patients (especially the elderly) [31,32].

The common pathway in all these pathological entities is vascular wall damage. Numerous studies in this direction followed, establishing the link between COVID-19 and vascular injury. The virus was shown to attack the endothelium of the vessels. As in the lungs, ACE2 is also expressed by the vascular wall, which logically leads to damage. Diseases of the lungs, heart, kidneys, and gastrointestinal organs are due to direct oxidative stress caused by the infection or are the result of an inadequate inflammatory response of the body [33].

Normally, blood vessels are coated by a monolayer of endothelial cells, which balance multiple mechanisms (both pro- and anticoagulant). Furthermore, Neubauer et al. demonstrated that endothelium regulates the interactions between the coagulation system and the surrounding cells [34].

The human body possesses an inbuilt system in which the blood remains fluid. However, injury to the blood vessel initiates thrombogenesis. Virchow describes three triggering events that, in themselves, may lead to the unlocking of a whole cascade of processes leading to thrombosis: endothelial injury, blood flow alteration, and hypercoagulability. It follows that endothelial cells have a leading role in the normal functioning of the cardiovascular system and, in particular, the maintenance of adequate hemostasis and coagulation [35]. An intact endothelium has a few essential functions: it protects the flowing blood from thrombogenic influence; it elaborates a thrombosis inhibitory factor, heparin-like substance, thrombomodulin, and inhibitors of platelets aggregation, and it releases thrombosis-favoring (prothrombic) factors, which means the endothelium maintains high blood flow and vascular relaxation in the normal state. Also, the endothelium exerts platelet inhibitory functions, fibrinolytic properties, etc. [34].

However, vascular injury exposes the subendothelial tissues, like collagen, elastin, fibronectin, and glycosaminoglycans, which are thrombogenic and are essential in initiating hemostasis and thrombosis [35]. When endothelial cells are damaged, there is a decreased production of vasorelaxants and antithrombotic mediators, and, on the other hand, the production of prothrombin factors and contraction are increased. Since the beginning of the pandemic, there has been a vast search for drugs that stop or reduce viral replication. The other goal was to prevent the overly aggressive immune response to the virus. After discovering the relationship between the infection caused by SARS-CoV-2 and endothelial damage, a search for suitable drugs to affect these disorders has also begun. The main well-studied and widely used medications used to treat ED are renin–angiotensin system (RAS) inhibitors and statins.

RAS inhibitors are a proven drug in the treatment of ED. A number of studies in the UK and Spain indicate a reduced risk in diabetic patients and fewer hospitalizations in patients on RAS inhibitor treatment. However, there are some concerns with COVID-19 infection, as RAS inhibitors increase the action of ACE2 receptors, which are the gateway for the virus. This could potentially increase the susceptibility of the endothelium to infection [35].

Statins are another medication that improve endothelial function in patients at risk. Statins reduce low-density lipoprotein (LDL) cholesterol, increase the expression of endothelial nitric oxide synthase (eNOS) expression, and suppress pro-oxidant enzymes [36]. Improvement of ED in rheumatoid diseases has been well studied, suggesting the involvement of statins in ameliorating changes in inflammatory diseases. In addition, statins positively affect patients with influenza [37].

In addition to drug treatment, it is particularly important to maintain a healthy lifestyle, including reasonable physical activity, a good diet, maintaining a normal weight, and quitting smoking. Additionally, optimal vitamin D levels may have a role in preventing endothelial dysfunction via the control of interferon 1-alpha [38]. All this supports a good condition of the cardiovascular system.

## 5. Pregnancy and COVID-19 Outcomes Associated with Endothelial Dysfunction

As mentioned above, any changes in the structure and function of the endothelium could cause numerous complications. This is especially valid during pregnancy. Although the pathophysiology of ED in pregnancy complications is not fully revealed, it is suggested that preeclampsia and fetal growth restriction are associated with ED [39].

The most typical pregnancy complication related to ED is preeclampsia. It could be preeclampsia based on primary ED or secondary ED based on primary impaired placental perfusion [40].

Hover, Rolnik DL commented on the Mendoza et al. discussion about the possibility of COVID-19 causing preeclampsia [5]. Rolnik discussed the high cumulative incidence of preeclampsia in pregnant COVID-19 patients and implicated that this should be interpreted cautiously due to the role of confounding factors [7]. Rolnik also suggested using ultrasound and biomarkers, such as the sFlt-1/PlGF ratio, to distinguish severe COVID-19 from preeclampsia in pregnant patients [7].

However, Espino-y-Sosa et al. later suggested a novel ratio in pregnant COVID-19 women—sFlt/angiotensin-II (ANG-II) [41]. This ratio was associated with adverse outcomes: severe pneumonia, ICU admission, intubation, sepsis, and death in pregnant women with COVID-19. As sFlt-1 causes ED and sensitizes the endothelial cells to ANG-II effects, probably the former biomarker contributes to the adverse COVID-19 related adverse outcomes [41].

Another mechanism of ED in preeclampsia is the activation of the immune system and increased secretion of cytokines, chemokines, and reactive oxygen species in the placenta [42].

Lambadiari et al. discussed the association of SARS-CoV-2 infection with impaired vascular function, altered endothelial glycocalyx, and myocardial deformations after COVID-19. In this case-control prospective study that included 70 patients, the authors demonstrated that residual cardiovascular symptoms four months after infection were associated with increased ED and oxidative stress biomarkers [43].

From a clinical point of view, the consequences of ED in preeclampsia are proteinuria, hypertension, and edema [39]. However, it is challenging to assess the degree of ED in preeclampsia. Along with the invasive and noninvasive methods for the evaluation of ED, the measurement of some markers for ED can be helpful: endothelin-1; vascular adhesive molecule 1 (VCAM-1); E-, P-, and L-selectin; thrombomodulin; endothelial glycocalyx degradation markers (endocan-1, hyaluronan, syndecan-1); von Willebrand factor; etc. [39]. Yinon et al. demonstrated that preeclampsia is associated with sleep-disordered breathing and ED. The authors speculated that preeclamptic toxemia and respiratory disturbances could contribute to blood vessel abnormalities and endothelial damage [44].

Speaking of ED and preeclampsia, Palomo et al. demonstrated differences and similarities in the endothelial and angiogenic profiles of pregnant women with preeclampsia and COVID-19 [45]. Using biomarkers of ED, coagulation, and innate immune response, such as VCAM-1, soluble tumor necrosis factor (TNF)-receptor I, heparan sulfate, von Willebrand factor antigen, α2-antiplasmin, C5b9, neutrophil extracellular traps, PlGF, sFlt-1, and angiopoietin 2, they demonstrated that both pregnant women with preeclampsia and COVID-19 showed altered profiles of circulating biomarkers compared to healthy pregnant women. The authors hypothesized that differentiation between COVID-19 and preeclampsia is challenging in some cases, suggesting both conditions activate inflammatory signaling pathways [45]. Here, we again have to emphasize that most of the biomarkers of ED should be interpreted along with other parameters while considering their limitations.

A case report by Naeh et al. described an otherwise healthy pregnant woman infected with SARS-CoV-2 who developed secondary hypertension, proteinuria, and liver dysfunction, forming a preeclampsia-like syndrome. The authors differentiated both conditions using normal PIGF testing, which ruled out preeclampsia with a very high negative predictive value [46].

Fabre et al. investigated hypertensive disorders of pregnancy, finding that SARS-CoV-2 is frequently isolated from the placenta in patients with gestational hypertension and preeclampsia. Moreover, the higher viral load was associated with more severe hypertension disorder, unrelated to the nasopharyngeal load [47].

Chornock et al. focused on the incidence of hypertensive disorders during pregnancy and concomitant COVID-19 to report the incidence of 34.2% of hypertension in COVID-19 pregnant women and 22.9% in pregnant women who tested negative for SARS-CoV-2 infection. After adjusting, the risk of developing hypertensive disorder during pregnancy if infected with SARS-CoV-2 was 1.58 [0.91–2.76] [48].

Gychka et al. reported that SARS-CoV-2-infected pregnant women suffer from placental vascular remodeling. The authors described 2-fold increased thickness and 5-fold decreased lumen area in placental arteries and smooth muscle proliferation and fibrosis. Taken together, these changes could be associated with complications during COVID-19 pregnancy [49].

Tou et al. also reported a case with overlapping presentation of hemolysis, elevated liver enzymes, and low platelet count (HELLP) syndrome and COVID-19. The authors described a 31-year-old woman who presented with hypertension, liver dysfunction, and hypercoagulability, which overlapped between HELLP syndrome and concomitant COVID-19 [50].

A Garcia Rodriguez et al. case report demonstrated that COVID-19 is a risk factor for preeclampsia and neurological manifestations. A 35-year-old pregnant and SARS-CoV-2- infected woman presented with tonic-clonic seizures with hypertension after a C-section. The authors concluded that COVID-19 promotes brain endothelial damage, leading to an increased risk of neurological complications during pregnancy [51].

Inflammation is considered a leading cause in neurovascular disorders’ acute and subacute phases. A reduction in inflammation has been shown in various studies to limit the damage caused by ischemic strokes, but the mechanism of action has not yet been studied in detail. There are MRI paradigms for diagnosing neuroinflammation that track vascular cell adhesion molecule-1 (VCAM-1), a molecule expressed in the blood vessel’s luminal side of the endothelium [52]. The implementation of this study is possible thanks to the application of microsized contrast agents. VCAM-1-tracking MRI demonstrates an “inflammatory penumbra” in ischemic infarcts. Damage to the parenchyma lasts up to 5 days after the appearance of the blow, and various inflammatory components are involved in the mechanism of the injury. In some patients, controlling inflammation has limited brain damage, thus showing the need for the molecular imaging of endothelial activation [52].

Fetal growth restriction could also be associated with ED during pregnancy. Yoshida et al. reported ED and hypoxia in the placenta related to increased sFlt-1 and the suppressed production of VEGF and PIGF [53], similar to Torres-Torres et al. [54].

McElwain et al. suggested that ED and vasculopathy observed in some obstetric complications, such as preeclampsia and gestational diabetes, are related to oxidative stressors, proinflammatory mediators, and interaction between adipose tissue and placenta [55].

The long-term effects of fetal growth restriction impact both the mother and the fetus [56]. The most important clinical consequence is the increased risk of chronic kidney disease in infants born after fetal growth restriction. Although it is unclear whether ED increased the risk of end-stage renal disease, decreased glomerular filtration, and increased albuminuria, ED and cardiovascular dysfunction are common complications in these pregnant women [57,58].

ED is a common feature of gestational diabetes. Moreover, endothelial damage caused by hyperglycemia and chronic low-grade inflammation is believed to increase mothers’ and fetuses’ morbidity and mortality [59].

Flores-Pliego et al. focused their research on the thrombotic and microvascular injury in the placental endothelium of pregnant women infected with SARS-CoV-2 [60]. The authors showed increased expression of von Willebrand factor in placental endothelium (decidua and villi) and decreased claudin-5 and vascular endothelial cadherin in women with severe COVID-19. These changes in the biomarkers were highly associated with leaky endothelium and thrombosis, which contribute to complicated pregnancy and complicated COVID-19 [60].

Noninvasive ventilation is a useful treatment alternative in patients who do not meet the criteria for intubation or when invasive ventilation is not available, especially in a pandemic situation when resources may be extremely limited. Positive pressure ventilation leads to elevated transalveolar pressure, which may cause alveolar rupture and air leakage into the extra-alveolar tissue. Gabirelli et al. report a specifically higher risk of pulmonary barotrauma in pregnant women treated with noninvasive ventilation [61].

Unfortunately, there is a lack of data regarding possible treatment options in pregnant women and SARS-CoV-2 infection due to the complex ethical aspects of such a study. However, there is slight evidence of the use of neutralizing monoclonal antibodies as a possible therapeutic agent during pregnancy. Frallonardo et al. report a series of 13 pregnant women with SARS-CoV-2 infection that were treated with neutralizing monoclonal antibody (Sotrovimab). The results revealed no adverse reactions and good clinical outcomes in all patients [62].

## 6. Pregnancy, Thrombophilia, and COVID-19

In this review, we focused on the endothelial dysfunction associated with pregnancy and COVID-19, and we were interested in whether there is a link between thrombophilia complications in pregnant women (with or without SARS-CoV-2 infection). The term “thrombophilia” is referred to as a deficiency of natural anticoagulants, mild prothrombotic mutations, and antiphospholipid antibodies, according to the International Society on Thrombosis and Haemostasis [63].

There are not many studies on the topic of COVID-19 in pregnant women with thrombophilia. However, Vuorio et al. discussed approaches for preventing further ED in pregnant women with SARS-CoV-2 infection and familial hypercholesterolemia, prone to myocardial infarction. The author suggested that treatment for familial hypercholesterolemia should be based on a case-by-case assessment, including statins [64].

Grandone et al. focused on COVID-19 pregnant women with alterations in the von Willebrand factor and ADAMTS13 axis. SARS-CoV-2 infection in these women was associated with endothelitis, endothelial damage, thrombotic microangiopathy, and an increased risk of preterm delivery [65].

A review article by Agostinis et al. paid attention to the relationship between COVID-19, preeclampsia, and the complement system [66]. The authors provided evidence for the crucial role of the complement system via its cross-talk with the coagulation system in the development of severe COVID-19. Through ACE2 and TMPRSS2 receptors, SARS-CoV-2 can provoke ED and the disruption of vascular integrity, causing hyperinflammation and hypercoagulation. Additionally, the suppression of ACE2 activity by the virus leads to bradykinin increase [66]. Thus, SARS-CoV-2-infected women are susceptible to developing complications related to ED, such as preeclampsia. However, due to the overlap of COVID-19 and preeclampsia, the treatment regimens are also shared, especially those targeting C3 or mannose-binding lectin (MBL)—the associated serine protease (MASP)-2.

It is well-documented that patients with COVID-19 are more prone to thrombotic complications (i.e., deep-vein thrombosis, pulmonary embolism, stroke, etc.) associated with multiorgan failure and increased mortality [67,68].

Some authors demonstrated evidence that hereditary thrombophilia is a risk for severe COVID-19 and a marker of poor prognosis. Thus, testing for hereditary and congenital thrombophilia may benefit patients with early diagnosis and prevention of thromboembolism [69,70].

Regarding pregnant COVID-19 patients with thrombophilia, data are scarce. However, whenever the SARS-CoV-2 infection is mild, moderate, or severe, the Royal College of Obstetricians and Gynaecologists (RCOG) recommends using thromboprophylaxis for pregnant women during hospital admission. Women already receiving LMWH prophylaxis or with a history of thrombophilia should continue treatment. Depending on their venous thromboembolism (VTE) score, they may also need an extended or higher dose prophylaxis. Additionally, according to the current risk stratification technique, which views COVID-19 infection as a temporary risk factor, a risk assessment should be conducted for afflicted women who do not need to be admitted. Longer-term thromboprophylaxis should be taken into consideration for female patients with persistent morbidity [71,72].

Speaking of the thromboprophylaxis of patients with COVID-19, timing and dosing are critical. Although pregnant women were excluded from the clinical trials, thrombophylactic treatment should be offered to pregnant women with thrombophilia and COVID-19. After considering the risks and benefits, and taking into account the significant bleed risk of between 0.9% and 1.9% of patients on therapeutic anticoagulants and thrombophilia, respectively, as well as thrombocytopenia, it could be associated with an increased mortality from SARS-CoV-2 infection [73,74].

Randomized-controlled trials showed that higher doses of low molecular weight heparin (LMWH) compared to the standard dosage failed to benefit severe COVID-19 [73]. However, the RCOG, in their Green-top Guideline No37 on reducing the risk of venous thromboembolism during pregnancy and puerperium recommended cases of heritable thrombophilia be managed using standard doses of thromboprophylaxis with the measurement of anti-Xa levels. However, women with previous VTE due to antithrombin deficiency should be treated with higher doses of LMWH.

Aspirin (75–150 mg) may be used to prevent preeclampsia, outweighing the risk of the adverse effects of SARS-CoV-2 infection [75].

Compared to aspirin alone, a combination of aspirin and LMWH may increase the rate of live births in patients with antiphospholipid syndrome and recurrent pregnancy loss [76]. However, enoxaparin in pregnant women (with or without thrombophilia) exhibited benefits, i.e., increased live births [77].

Furthermore, Daru et al. claim confidently that since the rate of VTE rises with the severity of COVID-19, if there are no heparin-related contraindications, thromboprophylaxis should be carried out in all COVID-19-symptomatic pregnant women until full recovery [78].

In contrast, other authors question thromboprophylaxis in pregnant women infected with SARS-CoV-2, especially for complications such as thrombophilia, preeclampsia, and intrauterine growth retardation [79].

They base their discussion on the lack of extensive clinical trials on preventing thrombosis. Thus, it is not possible to directly extrapolate data from nonpregnant patients with or without thrombophilia to pregnant patients and to put COVID-19 in the equation.

A systematic review and critical analysis of hemostatic and thrombo-embolic complications in pregnant women with COVID-19 conducted by Servante et al. showed that due to increased rates of coagulopathy and thromboembolism in pregnant COVID-19 patients, women should be risk-stratified and then be considered for thromboembolism prophylaxis [80].

To sum up, pregnant women are prone to a hypercoagulability state, which is further increased by concomitant thrombophilia and COVID-19, leading to an increased risk of thrombotic-related morbidity; thus, appropriate prophylaxis is recommended [81].

## 7. Healthcare Services Problems Regarding ED Management

The COVID-19 pandemic has significantly impacted healthcare systems, and the response to the epidemic has put further strain on medical professionals.

On the one hand, many doctors were not ready for an outbreak of this magnitude due to the pandemic’s unparalleled severity and exponential growth in cases around the globe, at least in comparison to the previous century. Several variables, including psychological stress and burnout, have severely impacted physicians’ well-being and may lead to increased medical mistakes and malpractice risk, negatively influencing patient care. Long working hours and decreased sleep quality over shifts due to lack of medical staff, workload requirements, inadequate compensation, interpersonal communication, and poor leadership have all historically been considered relevant factors [82]. Many doctors were forced to switch departments rapidly, frequently on short notice, and they suddenly found themselves working in a strange setting. Special medical equipment with barely breathable masks and protective clothing for the medical staff were among the factors that complicated the situation furthermore. They were forced to tackle fresh and unforeseen clinical issues while providing care for patients who were not their normal ones [83].

On the other hand, there were issues when people sought medical advice because the regular healthcare system was disturbed. Even if the COVID-19 disease does not directly increase maternal mortality relative to the general population, unsupervised pregnancies and the absence of regular antenatal appointments may indirectly negatively impact maternal health. Furthermore, the treatment of pregnant women was hampered by social constraints and loneliness. To some extent, telehealth was developed to help with the dilemma. Only a physical examination of the patient can diagnose some medical disorders. However, high-risk factors in pregnant women are neglected in the absence of routine treatment, which leads to severe consequences [84].

Thus, we can conclude that the COVID-19 pandemic, like other pandemics, poses a significant challenge to the healthcare sector due to its indirect effects on the vulnerable prenatal population and a rise in pregnancy-related problems. Along with healthcare services, patients must also be informed about the value of routine visits, safety measures like physical separation and mask use, and personal hygiene. While leaving the house should be avoided, health should not be put at risk in the process.

An overview of the studies on COVID-19 in pregnant women and registered coagulation disorders is presented in Table 1.

## 8. Conclusions

Endothelial dysfunction is a probable common pathophysiological mechanism in many complications related to COVID-19 and pregnancy, contributing to higher morbidity and mortality. Therefore, further research focusing on precise diagnostic methods and specific treatment targeted at ED may improve patient outcomes.

## Figures and Tables

**Table 1 biomedicines-11-02458-t001:** Literature review of pregnant COVID-19 patients with a coagulation disorder.

Authors	N °	Age	Time of the Diagnosis	Coagulopathy Type	Anticoagulation Therapy	Maternal Outcome	Fetus Outcome
Aminimoghaddam et al. [85]	1	21 y old	29 GW	TTP	Heparin	Alive	Preterm delivery in 29 GW; baby alive
Ballmick et al. [86]	1	18 y old	7 GW	Ischemic infarction in the right middle cerebral artery territory	Aspirin and clopidogrel	Alive	At follow-up, still pregnant with normal fetus parameters
Gunduz et al. [87]	1	22 y old	35 GW	Venous sinus thrombosis	Low molecular weight heparin	Alive	Healthy baby
Konstantinidou et al. [88]	165	35 (34–37.5)	35.5 GW (27.3–37.8)	Thrombophilia and SARS-CoV-2 placentitis	NR	162 alive	6 stillborn babies
Servante et al. [80]	13	NR	NR	3 arterial and/or venous thrombosis 7 DIC 3 coagulopathy	Low molecular weight heparin	11 alive	Alive
Arora et al. [89]	19	NR	37–40 GW	3 thrombophilia 9 hypertensive disorder	NR	8 with maternal respiratory compromise	7 patients with fetal distress syndrome
Mongula et al. [90]	1	27 y old	31 + 4 GW	Preeclampsia and DIC	Aspirin	Alive	Alive
Vlachodimitropoulou Koumoutsea et al. [91]	2	41 y old and 23 y old	35 + 3 GW and 35 + 2 GW	Progressive coagulopathy and transaminitis	Low molecular weight heparin	Alive	Alive

## Data Availability

Not applicable.

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
