# Peer review of "Endothelial Dysfunction and Pregnant COVID-19 Patients with Thrombophilia: A Narrative Review"

_biomedicines, 2023, doi:10.3390/biomedicines11092458_

Round 1
Reviewer 1 Report
I read the paper with great interest. I find it well written and with good research. The review is also well structured.
Below are my suggestions:
Introduction: update data on COVID-19 burden at the time of resubmission. Please add the information that other pandemics have had catastrophic consequences for pregnant women, such as H1N1, MERS, and SARS CoV1. And also, Malaria has worse consequences in pregnancy. I think it is important to underline pregnancy as a risk factor for severe illness due to COVID-19.
Methods: clear
Pregnancy and COVID-19 outcomes associated with endothelial dysfunction: well, add a limit to therapeutic options in pregnancy due to the lack of trials in pregnancy. The few data suggest the possible use of monoclonal antibodies (Use of Sotrovimab in a cohort of pregnant women with a high risk of COVID-19 progression: a single-center experience). Pathog Glob Health. 2023 Jul;117(5):513-519. doi: 10.1080/20477724.2023.2188839. and also there is a higher risk of barotrauma in pregnant women than in the general population (see and cite Barotrauma during non-invasive ventilation for acute respiratory distress syndrome caused by COVID-19: a balance between risks and benefits. Br J Hosp Med (Lond). 2021 Jun 2;82(6):1-9. doi: 10.12968/hmed.2021.0109. Epub 2021 Jun 30. PMID: 34191558.)
Conclusion: Give some public health proposals that come from your interesting review. In addiction, underline the consequences on maternal services due to the pandemic and the risk of burnout among health workers (see Risk of burnout and stress in physicians working in a COVID team: A longitudinal survey. Int J Clin Pract. 2021 Nov;75(11):e14755. doi:
10.1111/ijcp.14755. Epub 2021 Sep 8. PMID: 34449957; PMCID: PMC8646498
Author Response
Dear reviewers,
Thank you for your time to review our paper. We acknowledge that our paper might have some issues in conformity with the referees` comments. We have addressed them and revised the manuscript accordingly. Changes are visible as highlighted and/or track changes.
We sincerely thank the three reviewers for their thorough and helpful comments and suggestions. We have addressed all of the raised queries and responded to all reviewers' comments.
We believe that you find these changes satisfactory, and the revisions have substantially improved the quality of the manuscript.
I read the paper with great interest. I find it well written and with good research. The review is also well structured.
- Thank you very much for the overall evaluation of our paper as good. We acknowledge that our paper might have some issues in conformity with the referees` comments. We have addressed them and revised the manuscript accordingly.
Below are my suggestions:
Introduction: update data on COVID-19 burden at the time of resubmission.
- Thank you for the valuable note. We have updated the information regarding COVID-19 pandemic.
Please add the information that other pandemics have had catastrophic consequences for pregnant women, such as H1N1, MERS, and SARS CoV1. And also, Malaria has worse consequences in pregnancy. I think it is important to underline pregnancy as a risk factor for severe illness due to COVID-19.
- Authors’ reply: We appreciate your valuable remark. We have added and discussed the two topics suggested.
Methods: clear
Pregnancy and COVID-19 outcomes associated with endothelial dysfunction: well, add a limit to therapeutic options in pregnancy due to the lack of trials in pregnancy.
- Authors’ reply: Thank you very much for your useful suggestion. We agree that it is important to note the limitation of the paucity of data available regarding treatment options in pregnant women with COVID-19. The following text has been added to the “Pregnancy and COVID-19 outcomes associated with endothelial dysfunction” paragraph:
“Unfortunately, there is paucity of data regarding possible treatment options in pregnant woman and SARS-CoV-2 infection due to the complex ethical aspects of such study. “
The few data suggest the possible use of monoclonal antibodies (Use of Sotrovimab in a cohort of pregnant women with a high risk of COVID-19 progression: a single-center experience). Pathog Glob Health. 2023 Jul;117(5):513-519. doi: 10.1080/20477724.2023.2188839) and also there is a higher risk of barotrauma in pregnant women than in the general population (see and cite Barotrauma during non-invasive ventilation for acute respiratory distress syndrome caused by COVID-19: a balance between risks and benefits. Br J Hosp Med (Lond). 2021 Jun 2;82(6):1-9. doi: 10.12968/hmed.2021.0109. Epub 2021 Jun 30. PMID: 34191558.)
- Authors’ reply: Thank you for the valuable comment. We have added and discussed the two studies suggested. The following texts has been was added to the “Pregnancy and COVID-19 outcomes associated with endothelial dysfunction” paragraph of the manuscript:
“Non-invasive ventilation is a useful treatment alternative in patients who do not meet criteria for intubation or when invasive ventilation is not available, especially in a pandemic situation when resources may be extremely limited. Positive pressure ventilation leads to an elevated transalveolar pressure, that may cause alveolar rupture and leakage of air into the extra-alveolar tissue. Gabirelli et al., report a specifically higher risk of pulmonary bauratrauma in pregnant women treated with non-invasive ventilation. (Gabrielli M, Valletta F, Franceschi F; Gemelli Against COVID 2019. Barotrauma during non-invasive ventilation for acute respiratory distress syndrome caused by COVID-19: a balance between risks and benefits. Br J Hosp Med (Lond). 2021 Jun 2;82(6):1-9. doi: 10.12968/hmed.2021.0109. Epub 2021 Jun 30. PMID: 34191558.)”
“However, there is a small evidence of the use of neutralizing monoclonal antibodies as a possible therapeutic agent during pregnancy. Frallonardo et al, report a series of 13 pregnant women with SARS-CoV-2 infection that were treated with neutralizing monoclonal antibody (Sotrovimab). The results revealed no adverse reactions and good clinical outcome in all patients. (Frallonardo L, Vimercati A, Novara R, Lepera C, Ferrante I, Chiarello G, Cicinelli R, Mongelli M, Brindicci G, Segala FV, Santoro CR, Bavaro DF, Laforgia N, Cicinelli E, Saracino A, Di Gennaro F. Use of Sotrovimab in a cohort of pregnant women with a high risk of COVID 19 progression: a single-center experience. Pathog Glob Health. 2023 Jul;117(5):513-519. doi: 10.1080/20477724.2023.2188839. Epub 2023 Mar 10. PMID: 36896940; PMCID: PMC10262799.)”
Conclusion: Give some public health proposals that come from your interesting review. In addiction, underline the consequences on maternal services due to the pandemic and the risk of burnout among health workers (see Risk of burnout and stress in physicians working in a COVID team: A longitudinal survey. Int J Clin Pract. 2021 Nov;75(11):e14755. doi:
10.1111/ijcp.14755. Epub 2021 Sep 8. PMID: 34449957; PMCID: PMC8646498
- Authors’ reply: The referee is absolutely right to point this out. Thus, we have added this important information, but we have put it in a new paragraph of the manuscript: “Health care services problems”. You can track the changes with TrackChanges. modified the conclusion part.
Reviewer 2 Report
I have read with interest the review by Sekulovski et al, entitled “Endothelial dysfunction and pregnant COVID-19 patients with 2
thrombophilia: a narrative review”. The authors discuss properly the different aspects od Endothelial damage in general and in CoV19 pregnant women and in the complications related or not to viral infections. I have only a few points to be addressed to the authors.
1 The term “Thrombophilia” is referred to the meaning assigned by the ISTH, i.e., deficiency of natural anticoagulants, mild prothrombotic mutations, and anti-phospholipid antibodies (Franchini M, Martinelli I, Mannucci PM. Uncertain thrombophilia markers. Thromb Haemost. 2016 Jan;115(1):25-30. doi: 10.1160/TH15-06-0478.). In other words, the concept of thrombophilia should be considered from the Haemostasis and Thrombosis side! The other conditions are, in general, metabolic and haemodynamic risk factors for arterial thrombosis. Is there a link between thrombophilia and pregnancy complications with or without CoV19 infection? The authors should try to answer this question.
2 The pathophysiology of the endothelial involvement during CoV 19 infection and in general should be written more precisely (Neubauer K, Zieger B. Endothelial cells and coagulation. Cell Tissue Res. 2022 Mar;387(3):391-398, open access).
3 Searching for sophisticated biomarkers is not appropriate since it would very difficult to implement them in a non-specialized laboratory of a general hospital. I realize, however, that it could be of help in research projects.
4 Is there a role for anticoagulants (LMWHs) in pregnancy infected by CoV19? The authors do not consider this point which I think is important.
5 4. Endothelial dysfunction and COVID-19, line 185. The terms “prothrombin factors” are not adequate, “Tissue factor” is to be used instead (see point 2).
I have not recommendations on the English style.
Author Response
Pregnancy with SARS-CoV-2 infection can raise the risk of many complications, including severe COVID-19 and maternal-fetal adverse outcomes. Additionally, endothelial damage occurs as a result of direct SARS-CoV-2 infection as well as the immune system, cardiovascular and thrombo-inflammatory reactions. In this narrative review, we focus on endothelial dysfunction (ED) in pregnancy, associated with obstetric complications, such as pre-eclampsia, fetal growth retardation, gestational diabetes, etc., and SARS-CoV-2 infection in pregnant women that can cause ED itself and overlap with other pregnancy complications. We also discuss some shared mechanisms of SARS-CoV-2 pathophysiology and ED.
- Thank you very much for the overall evaluation of our paper as good. We acknowledge that our paper might have some issues in conformity with the referees` comments. We have addressed them and revised the manuscript accordingly.
Round 2
Reviewer 2 Report
I did not find what I have suggested. I ask the authors to reply to my review.
Author Response
I did not find what I have suggested. I ask the authors to reply to my review.
- We have to disclose that, somehow, at the moment of first revision, the only available information in the system from the second reviewer, was the previously copied and replied by us text. We apologize for the inconvenience and want to assure you that we have not missed your comments on purpose.
- Thank you for your time to review our paper. We acknowledge that our paper might have some issues in conformity with the referees` comments. We have addressed them and revised the manuscript accordingly. All new changes are visible as highlighted in green and track changes.
- We sincerely thank the reviewers for their thorough and helpful comments and suggestions. We have addressed all of the raised queries and responded to all reviewers' comments.
- We believe that you find these changes satisfactory, and the revisions have substantially improved the quality of the manuscript.
I have read with interest the review by Sekulovski et al, entitled “Endothelial dysfunction and pregnant COVID-19 patients with 2 thrombophilia: a narrative review”. The authors discuss properly the different aspects od Endothelial damage in general and in CoV19 pregnant women and in the complications related or not to viral infections. I have only a few points to be addressed to the authors.
- Thank you for the overall evaluation of our paper as good. We are thankful for the critical noted and we are addressing all of them.
1 The term “Thrombophilia” is referred to the meaning assigned by the ISTH, i.e., deficiency of natural anticoagulants, mild prothrombotic mutations, and anti-phospholipid antibodies (Franchini M, Martinelli I, Mannucci PM. Uncertain thrombophilia markers. Thromb Haemost. 2016 Jan;115(1):25-30. doi: 10.1160/TH15-06-0478.). In other words, the concept of thrombophilia should be considered from the Haemostasis and Thrombosis side! The other conditions are, in general, metabolic and haemodynamic risk factors for arterial thrombosis. Is there a link between thrombophilia and pregnancy complications with or without CoV19 infection? The authors should try to answer this question.
- The referee is right to point out the definition of thrombophilia according to ISTH (we cited the suggested paper). In the paper, we did our best to discuss thrombophilia from the mentioned aspects.
- We searched for information regarding the link between thrombophilia and pregnancy complications with/without COVID-19.
2 The pathophysiology of the endothelial involvement during CoV 19 infection and in general should be written more precisely (Neubauer K, Zieger B. Endothelial cells and coagulation. Cell Tissue Res. 2022 Mar;387(3):391-398, open access).
- Thank you for the valuable suggestion. We add more on the pathophysiology of the endothelial involvement during COVID-19, based on the paper of Neubauer et al. in section 4.
3 Searching for sophisticated biomarkers is not appropriate since it would very difficult to implement them in a non-specialized laboratory of a general hospital. I realize, however, that it could be of help in research projects.
- We appreciate the point and discuss further on this matter in the text (section 3 and section 5).
4 Is there a role for anticoagulants (LMWHs) in pregnancy infected by CoV19? The authors do not consider this point which I think is important.
- Thank you for the valuable suggestion. We add a passage on the topic, since it is essential for the topic. We added the information in section 6, after discussing the burden of thrombophilia on COVD-19 severity in pregnant women.
5 4. Endothelial dysfunction and COVID-19, line 185. The terms “prothrombin factors” are not adequate, “Tissue factor” is to be used instead (see point 2).
- Thank you for the note, we have changed the terms.
I have not recommendations on the English style.
- Thank you.
Round 3
Reviewer 2 Report
I have eventually read the emended form of this paper. What I asked has been satisfactory done by the Authors. At page 10, first line the term "risk" is lacking. Please check.
I have no significant remarks on the English form, except some minor mistakes.